# The Genome-Wide Analysis of RALF-Like Genes in Strawberry (Wild and Cultivated) and Five Other Plant Species (Rosaceae)

**DOI:** 10.3390/genes11020174

**Published:** 2020-02-06

**Authors:** Hong Zhang, Xiaotong Jing, Ying Chen, Zhe Liu, Yuting Xin, Yushan Qiao

**Affiliations:** Laboratory of Fruit Crop Biotechnology, College of Horticulture, Nanjing Agricultural University, Nanjing 210000, China; zhangh_19901025@163.com (H.Z.); 2018204005@njau.edu.cn (X.J.); cy315406@163.com (Y.C.); 2016204008@njau.edu.cn (Z.L.); xinyuting1989@njau.edu.cn (Y.X.)

**Keywords:** RALF, Rosaceae species, identify, classify, evolution, expression

## Abstract

The rapid alkalinization factor (RALF) gene family is essential for the plant growth and development. However, there is little known about these genes among Rosaceae species. Here, we identify 124 RALF-like genes from seven Rosaceae species, and 39 genes from *Arabidopsis*, totally 163 genes, divided into four clades according to the phylogenetic analysis, which includes 45 mature RALF genes from Rosaceae species. The YISY motif and RRXL cleavage site are typical features of true RALF genes, but some variants were detected in our study, such as YISP, YIST, NISY, YINY, YIGY, YVGY, FIGY, YIAY, and RRVM. Motif1 is widely distributed among all the clades. According to screening of cis-regulatory elements, GO annotation, expression sequence tags (EST), RNA-seq, and RT-qPCR, we reported that 24 *RALF* genes coding mature proteins related to tissue development, fungal infection, and hormone response. Purifying selection may play an important role in the evolutionary process of *RALF-like* genes among Rosaceae species according to the result from ka/ks. The tandem duplication event just occurs in four gene pairs (*Fv-RALF9* and *Fv-RALF10*, *Md-RALF7* and *Md-RALF8*, *Pm-RALF2* and *Pm-RALF8*, and *Pp-RALF11* and *Pp-RALF14*) from four Rosaceae species. Our research provides a wide overview of RALF-like genes in seven Rosaceae species involved in identification, classification, structure, expression, and evolution analysis.

## 1. Introduction

The rapid alkalinization factors (RALFs) belong to the small and cysteine-rich secreted peptides, which are involved in various processes of growth and development and play an important role in communication between cells in plants [1]. Generally, the molecular weight of RALF peptides ranges from 80 kDa to 90 kDa, except for a few special peptides which are larger [1]. Pearce et al. [2] showed that mature peptides were almost 50 aa and contain four highly conserved cysteines in the C-terminus, which are essential for activation and are likely involved in disulfide bridges. The mature peptides also contain other conserved regions, such as the YISY motif near the N-terminus, a GASYY motif between the first and second conserved cysteines, and a PYXRGCS motif that contain the third conserved cysteine residue [3]. The YIXY motif is the key regulatory functional element required for targeting RALF to its putative receptor. Meanwhile, when isoleucine (I) is replaced by alanine (A), this can cause a reduction in the alkalinization of suspension cells [2]. Campbell and Turner [4] identified 765 RALF proteins from 51 plant species and divided them into four clades. There are typical features of RALFs in clades I, II, and III, including the RRXL cleavage site and the YISY motif. Nevertheless, clade IV lacks the typical RALF trait, whereas the members of clade IV exhibit diverse expression profiles and physiochemical properties. As such, they hold that the peptides of clade IV are not the true RALF genes, and are described as the RALF-like peptides. Collectively, the release of mature RALF peptides is important to functioning.

RALF was initially discovered in the medium of tobacco suspension cells during the peptide screen for systemin proteins (defense-related, alkalinization-inducing peptides) [2]. Homologs of RALF have already been found in the same plant kingdom; for example, five homologs were found in poplar [5], 39 homologs of RALF were found in *Arabidopsis thaliana* [6], and 12 of these genes from the expression sequence tag (EST) database. RALFs have been identified in a variety of species including monocots, eudicots, and early diverging lineages [2,5,7,8,9,10,11], and Campbell and Turner [4] identified 765 RALF proteins from 51 plant species.

According to previous studies, *RALF*s have wide expression profiles, which vary based on tissue growth stage, biotic, or abiotic treatments. In *Arabidopsis thaliana*, *RALF*-*like* genes showed diverse expression profiles in different organs [12], as well as tissue- and stage-specific expression [5]. *RALF-like* genes also respond to nematode, drought [13], nitrosative, and oxidative stresses [11]. In particular, Mecchia et al. [13] reported that RALF4/19 peptides interact with LRX proteins to control pollen tube growth in *Arabidopsis*. Ge et al. [14] also reported that RALF4/19 peptide ligands enhance the strength of LLG2/3B–UPS/ANX interaction to influence the integrity of pollen tube. Moussu et al. [15] reported the crystal structure of the LRX-RALF cell wall complex and demonstrated that RALF peptides were activated as folded proteins, and RALFs can also instruct LRX cell wall modules and CrRKL1L receptors through structurally different binding modes to coordinate pollen tube integrity. *ScRALF23* was reported as able to affect the pollen mitosis by functioning among the process from sporophyte to gametophyte signaling [16]. Leucine-rich repeat extension proteins can combine with RALF to modify cell wall expansion and bind to the transmembrane receptor FERONIA and regulate the cell growth [17]. Haruta et al. [18] reported that *PtdRALF1* and *PtdRALF2* were continuously and stably expressed in several organs, whereas *PtdRALF2* showed low expression in the leaves of mature poplar trees, and *PtdRALF1* was hardly affected by phytohormones, nutrient concentration pathogen elicitors, and decreasing pH caused by HCl, but the expression of *PtdRALF2* could be inhibited strongly by MeJA after 5 h. Gupta et al. [19] reported that the expression of *RALF* in resistant chickpeas was induced by *Fusarium oxysporum* f. sp. *ciceri* (Race 1, Foc 1), a plant pathogenic fungus, but showed low expression levels in the susceptible chickpea cultivars. In strawberry, the authors of [20] reported that a *RALF*-*like* gene in *Fragaria* × *ananassa* decrease the fungal pathogen resistance of unripe strawberry fruit stages. *RALF23* can induce the FER to stabilize MYC2 and promote the JA-dependent response [21]. LRX1, the main LRX protein of root hairs, can interact with FER and RALF1 to coordinate growth processes [17]. *RALF* genes function extensively, but the researches in Rosaceae species are little, some evidences have been provided that RALF will influence the development of pollen tube and can respond to the fungal infection. Here, some Rosaceae species need cross-pollination and have severe fungal diseases, which may have relationship with *RALF*.

In the present study, we performed a genome-wide analysis to identify RALF-like genes of seven Rosaceae species (peach, apple, wild strawberry, cultivated strawberry, European pear, sweet cherry, and Japanese apricot). The analysis revealed that the seven Rosaceae species contain 124 RALF-like genes that were classified into four clades using phylogenetic analysis. Here, we selected 45 mature RALF genes from Rosaceae species, which include the RRXL cleavage site, the YISY motif, and four highly conserved cysteines in the C-terminus. Further, we analyzed the protein structure, duplication events, and evolutionary selections. Importantly, we combined cis-regulatory elements analysis, gene ontology (GO) annotation, EST data, RNA-seq analysis, and RT-qPCR results to have a clear knowledge of these genes in some development stages, tissues, and under diverse stresses. Specifically, we analyzed the coding capacity of these genes, and picked out 24 coding mature RALF genes, which can be further investigated. The research provides us with a comprehensive knowledge of RALF-like genes in seven Rosaceae species.

## 2. Materials and Methods

### 2.1. Data Sources

The peach (*Prunus persica*) genome (v2.0.a1) [22], apple (*Malus* × *domestica*) genome (HFTH1 Whole Genome v1.0) [23], wild strawberry (*Fragaria vesca*) genome (v4.0.a1) [24], cultivated strawberry (*Fragaria* × *ananassa*) genome (v1.0.a1) [25], European pear (*Pyrus communis*) genome (v1.0) [26], sweet cherry (*Prunus avium*) genome (v1.0.a1) [27], and related annotation files were downloaded from GDR database [28].

The genome and related annotation files of Japanese apricot (*Prunus mume*) (v1.0) [29] were retrieved from NCBI (https://www.ncbi.nlm.nih.gov/). The *Arabidopsis* RALF-like proteins were downloaded from the Phytozome database [30] using “RALF” as the keyword, and the RALF genes reported in another research were added [4]. The RNA-seq data for investigating the expression profiles of RALF-like genes were retrieved from European Nucleotide Archive (https://www.ebi.ac.uk/ena).

### 2.2. Identification and Sequence Analysis of RALF-Like Proteins

To identify the candidate members of the RALF-like proteins in seven Rosaceae species, the RALF domain (PF05498) was downloaded from the Pfam database [31], and HMMER v3.0 [32] was used to search the conserved domain sequences. Finally, these candidate sequences were submitted to Pfam [31] to confirm whether they had the RALF domain.

The predicted subcellular localization of candidate proteins was retrieved from Plant-mPLoc [33,34,35] (http://www.csbio.sjtu.edu.cn/bioinf/plant-multi/). Verified RALF-like protein sequences were submitted to the ExPASy proteomics service [36,37] to screen the essential information of proteins, and the signal peptide information was obtained from SignalP-5.0 [38].

### 2.3. Maximum Likelihood Phylogeny of RALF-Like Proteins

The protein sequences of the seven Rosaceae species and *Arabidopsis thaliana* were aligned using Muscle [39]. The model was calculated by using ModelFinder (VT+R6) [40]. Then, a phylogenetic tree was inferred under maximum likelihood with IQ-TREE [40,41]. The resulting tree file was visualized with FigTree version 1.4.4 (http://tree.bio.ed.ac.uk/software/figtree/) and then submitted to iTOL [42] (https://itol.embl.de/) for modification.

For future understanding of the function among 51 mature RALF genes, we selected the RALF genes with four conserved cysteine residues, YISY motif, and RRXL cleavage site to align, and build the phylogenetic tree (Best-fit model: JTT+I+G4) following the methods mentioned above.

### 2.4. The Conserved Domain and Motif Analysis

RALF-like proteins of peach, apple, wild strawberry, cultivated strawberry, European pear, sweet cheery, Japanese apricot, and *Arabidopsis* were aligned using Jalview version 2 (muscle with defaults) [43] to analyze the conserved regions and verify the RALF-like proteins twice.

In this research, six previously identified motifs [9] were submitted to the MEME suite (http://meme-suite.org/) [44], and the FIMO (Find Individual Motif Occurrences) service was used to screen among RALF-like proteins.

### 2.5. The Coding Capacity of RALF-Like Genes in the Seven Rosaceae Species

All the RALF-like coding sequences (CDs) from the seven Rosaceae species were submitted to coding potential calculator (CPC) (Center for Bioinformatics, Peking University, China) (http://cpc.cbi.pku.edu.cn/programs/run_cpc.jsp) [45] to analyze their coding abilities.

### 2.6. Duplication Event Analysis and ka and ks Values among RALF-Like Genes

Potential homologous gene pairs were identified in each of the genomes of the seven Rosaceae species using BLASTP [46] (*E* < 1·10^-5^, top 5 matches). The results of BLASTP and the arranged CFF files were then used as the input files of Mscanx [47] to analyze the duplication events. The RALF-like CDs sequences of *Arabidopsis* and seven Rosaceae species were translated to proteins in MEGA7 [48] and aligned using ClustalW, then we submitted the aligned CDs sequences to DnaSPv6.12.03 [49] to calculate the ka and ks values.

### 2.7. The Expression Analysis Using Expression Sequence Tag Data

The EST databases of strawberry, apple, and peach (only these three species’ EST data can be found) were downloaded from PlantGDB (http://www.plantgdb.org/) [50]. A local BLASTN (*E*-value < 1·10^−5^, top 5) was performed against pear EST libraries to get the hits for each *RALF*-*like* gene.

### 2.8. Gene Ontology and cis-Regulatory Elements Analysis

All 163 RALF-like proteins were submitted to Blast2GO [51] to obtain the gene ontology annotation information, with all the options as default. The 2 kb region upstream of the transcriptional start site was used for cis-regulatory elements analysis with the Plant Care [52] online service (http://bioinformatics.psb.ugent.be/webtools/Plantcare/html/), and the result was depicted using TBtools [53].

### 2.9. The Expression Profiles of RALF-Like Genes in Strawberry and Apple from RNA-Sequencing

To analyze the wide expression profiles, RNA-seq data were used, and the FASTQ files were downloaded from European nucleotide archive (https://www.ebi.ac.uk/ena). Here, *Fragaria vesca*, *Fragaria* × *ananassa*, and *Malus* × *domestica* (only these three species in the present study can be found the available RNA-seq data to analyze the expression profiles about diverse tissues and stresses) were considered when investigating the RALF-like gene expression of diverse tissues, organs, developmental stages, and stresses. The study accessions from NCBI were SRP096282, SRP098567, ERP004230, SRP035308, SRP018410, ERP013896, SRP111905, SRP125281, SRP091754, and SRP139480 [54,55,56,57,58,59,60,61], which were involved in the development of flower tissues, fruit tissues, seed tissues, and fungal stresses. Ultimately, the FPKM (fragments per kilobase per million) [62] of RALF-like genes was extracted from the GTF files and submitted to TBtools [52] to draw the heatmaps with the log_2_ translation of FPKM.

### 2.10. Gene Validation Experiments by Quantitative Real-Time PCR (RT-qPCR)

*F.* × *ananassa* cv. Benihoppe plants cultured in MS (Murashige and Skoog) solid medium for four weeks to perform, and 0.2 mM NAA (naphthylacetic acid), 0.2 mM ABA (abscisic acid), 0.2 mM GA_4_ (gibberellin 4), 0.2 mM MeJA (methyl jasmonate), and 1.0 mM SA (salicylic acid) were sprayed on the tissue of cultured seedlings. NAA, ABA, and GA_4_ will influence the growth of plants, and MeJA and SA are related to the stresses of plants. Three plants were used for one treatment, and the leaves of strawberry seedlings were immediately flash frozen using liquid nitrogen and stored in a −80 °C freezer after treating for 12, 24, and 48 h.

Total RNA was isolated according to the instructions of the RNA extraction kit (Tiangen, Beijing, China), and RNA was reverse-transcribed into complementary DNA (cDNA) using the PrimeScript RT reagent kit (TaKaRa, Dalian, China) for quantitative real-time PCR (RT-qPCR). RT-qPCR was performed on an ABI 7300 Real-Time PCR System (Applied Biosystems, Foster City, CA, USA). The gene-specific primers were designed by the Beacon Designer 7 program (Premier Biosoft, Palo Alto CA, USA). *EF-1α* (XM_004307362) was used as an internal control to normalize the expression level of target genes. All primers are listed in Table 1. The cycling conditions were maintained as follows; 95 °C for 4 min, 40 cycles at 95 °C for 25 s, 60 °C for 20 s, and 73 °C for 43 s to create a melting curve. The reaction was performed in a 20 µL reaction mixture including a diluted cDNA sample as a template, a SYBR premix ExTaq (2×) (TaKaRa, Kyoto, Japan), and primers. The relative expression level of genes was calculated by the 2^−∆∆CT^ method [63].

## 3. Results

### 3.1. Identification, Coding Capacity, Classification, and Structural Features of RALF-Like Genes in Rosaceae Species

The genome-wide search for and verification of the seven Rosaceae species reference genomes indicated that there are 124 sequences with the RALF domain, with 41 from cultivated strawberry, 13 from wild strawberry, 20 from apple, nine from Japanese apricot, 17 from sweet cherry, 14 from peach, and 10 from European pear, in addition to 39 *Arabidopsis* RALF-like protein sequences that were retrieved from Phytozome [23] and previous research [4], among these 163 sequences, there are five common protein pairs (Pp-RALF1 and Pp-RALF2, Fv-RALF5 and Fa-RALF10, Fa-RALF7 and Fv-RALF6, Fa-RALF12 and Fv-RALF10, and At-RALF7 and At-RALF37).

The RALF-like proteins’ primary structure information is listed in Appendix A. The molecular weight ranges from 7 to 121 kDa, the number of amino acids ranges from 64 to 747, and the minimum theoretical isoelectric point (pI) is 4.42 and the maximum is 10.31. In the seven Rosaceae species, there were 81% (101/124) RALF-like proteins with a pI higher than 7, and others lower than 7. Therefore, most are basic amino acids. There are ~86% (107/124) of these proteins which have a signal peptide, which is essential for secreted proteins. The majority of RALF-like proteins located extracellularly are also involved in other positions, such as cell membrane, cytoplasm, endoplasmic reticulum, mitochondrion, and the nucleus, which implies these proteins mainly function outside the cell.

The results of CPC [45] showed that *Fa*-*RALF22* and *Fa*-*RALF23* have no coding capacity, and 32 have a weak noncoding capacity, 58 have the weak coding ability, and 32 have a clear coding capacity (Appendix A).

According to the phylogenetic analysis (Figure 1), these RALF-like proteins were divided into four clades (I–IV), and clade I was divided into nine subclades (A–G, S1, and S represents the single sequence) and clade III was divided into two subclades (A and B). The majority (~57% (71/124)) of RALF-like proteins in Rosaceae species belong to clade I, 24 RALF-like proteins to clade II, 26 to clade III, and three to clade IV. Clade I–D, clade I–F, clade II, and clade III-B contain all the Rosaceae species, clade I–A contains five Rosaceae species, and clade I–C and clade I–E contain six Rosaceae species. Clade I–G and clade III–A contain four Rosaceae species. Others contained one Rosaceae species, which implied that most RALF-like proteins identified in the present study have conserved functional structures.

Combining the alignment from Jalview Version 2 (muscle with defaults) [43] (Appendix A) and the analysis of motifs retrieved from MEME [44] (Appendix A), we found some structural features among clades or subclades according to the alignment and motif results.

A majority of the RALF-like proteins have four conserved cysteines, except 22 sequences from the Rosaceae species (sequences marked with green background in Appendix A). There are 51 sequences which have the YISY motif, the RRXL cleavage site, and four conserved cysteine residues the typical features of mature RALF genes and 45 from the Rosaceae species. All the sequences in clade I-F and most sequences of clade I–D, clade I–E, and clade II (seven from Rosaceae species in Clade I–D, eight from Rosaceae species in Clade I–E, 18 from Rosaceae species in Clade II) have typical features. These mature RALF-like genes may be more meaningful for future research, whereas other sequences have none or one of these features. Searching these sequences, some YISY motifs may have some mutations, such as YISP, YIST, NISY, YINY, YIGY, YVGY, FIGY, YIAY, and RRXL, which may mutate to RRVM.

These protein sequences were searched for the presence of six motifs (Figure 2), and motif1 was found to exist in 141 sequences and 110 sequences from Rosaceae species. The second and third most common motifs are motif3 (84 sequences with 71 from Rosaceae species) and motif2 (75 sequences with 63 from Rosaceae species). Motif1 distributed widely among all the clades.

As depicted in the phylogenetic tree of 51 mature RALF genes (Figure 3), we selected 10 homologous genes pairs (Md-RALF4 and Pc-RALF2; Pp-RALF5, Pm-RALF5, and Pa-RALF10; Fa-RALF5, Fa-RALF29, and Fa-RALF35; Pm-RALF7 and Pp-RALF6; Fa-RALF2, Fv-RALF2, and Fa-RALF40; Md-RALF14 and Md-RALF5; Pp-RALF12 and Pa-RALF3; Pm-RALF6, Pp-RALF8, and Pa-RALF2; Pc-RALF9 and Md-RALF15; Fa-RALF19, Fv-RALF1, Fa-RALF33, Fa-RALF24, and Fa-RALF14) to analyze.

### 3.2. Chromosome Location, Duplication Events, and Divergence Rates

As depicted (Appendix A), there is an expanding distribution among Rosaceae species according to the chromosomal location results. In wild strawberry, there are six chromosomes, except Fvb7, in which 13 RALF-like genes were located. In cultivated strawberry, there are 21 chromosomes which have RALF-like gene sites. The following numbers of RALF-like genes were identified in specific chromosomal locations: 10 in apple, 8 in sweet cherry, 6 in peach, and 9 in European pear. In Japanese apricot, which has eight chromosomes, Pltd and Unplaced Scaffold regions, RALF-like gene sites were found on six chromosomes, whereas *Pm-RALF9* is located in Un region. As shown, most of the RALF-like genes are distributed around the telomere, and others around the centromere.

Genes can be duplicated by numerous mechanisms, including through whole genome duplication or segmental (WGDs), dispersed gene duplication (DD), tandem duplication (TD), proximal duplication (PD), and singleton (single-copy, SC) events. Among the 124 sequences, 45.2% (56/124) involved WGD events, 33.1% (41/124) dispersed gene duplication events, 9.7% (12/124) tandem events, 5.6% (8/124) proximal duplication events, and 6.4% (7/124) singleton events (Appendix A). Only four couples of tandem duplication genes, *Fv-RALF9* and *Fv-RALF10*, *Md-RALF7* and *Md-RALF8*, and *Pm-RALF2*, and *Pm-RALF8*, *Pp-RALF11*, and *Pp-RALF14* were found (Appendix A).

The ka and ks were calculated by DnaSP [49] using all the RALF-like CDs sequences identified. As is calculated, the ka range is 0 to 4.8, ks range is 0 to 4.1, and the ka/ks is 0 to 22.1. The data distribution map was displayed through box plots which were constructed via Origin (OriginLab Corporation, Northampton, MA, USA) and the outlines were eliminated automatically (Figure 4). The ka and ks values were stably distributed (Figure 4a,b), which implies there were stable evolutionary counts and mutation frequencies between and among these species. The ka/ks orients the evolution direction. As depicted (Figure 4c), all the average mean values of genes pairs were lower than one, which implies that purifying the selection may promote evolution. The values of ka, ks are listed in Appendix A.

### 3.3. Gene Ontology and Cis-Regulatory Elements Analysis

Among 124 sequences (RALF-like genes in *Arabidopsis* not included), 17 GO terms were found in 51 RALF-like genes. Approximately 85% (138/163) of RALF-like genes in the present study (conclude the sequences from *Arabidopsis thaliana*) were enriched in the GO category of cellular component (CC). The gene number of each GO term and functional class were counted and depicted in Figure 5. The GO categories of the biological process (BP) and molecular function (MF) accounted for 74% (120/163) and 33% (54/163), respectively. The calcium-mediated signaling (GO: 0019722), plasmodesma (GO: 0009506), protein phosphorylation (GO: 0006468), hormone activity (GO: 0005179), cell–cell signaling (GO: 0007267), and the integral component of the membrane (GO: 0016021) are major GO categories.

There are 21 cis-regulatory elements in total, such as abscisic acid (ABA), anaerobic (Ana), auxin (Auxin), cell cycle (Cyc), circadian control (Cir), defense and stress (DS), differentiation of the palisade mesophyll cells (DPMC), drought (Drought), endosperm specific expression (Endosperm), flavonoid biosynthetic (Fla), gibberellin (Gibberellin), light (Light), low temperature-related (LT), methyl jasmonate (MeJA), meristem-specific expression (Meristem), phytochrome downregulation-related (Phy), (Root), salicylic acid (SA), seed-specific expression (Seed), wound-related (Wound), and zein metabolism-related (Zein), which are involved in stress and tissue-specific expression (Appendix A and Appendix A). The major elements contain light-related elements, with 998 counts, followed by anaerobic (256), abscisic acid (195), drought (112), and gibberellin (95). The quantity of every RALF-like gene ranges from 0 to 12 (Appendix A).

### 3.4. The Expression Information from Expression Sequence Tag Data

As listed in Appendix A, according to the results from ESTs, we reported that twelve plant tissues (flower, bud, fruit, fruit mesocarp, fruit core, fruit cortex, phloem, xylem, leaf, shoot internodes, young root, and young shoot), six abiotic stresses (cold, water, drought, heat, and salt), and two biotic stresses (*Venturia inaequalis* and *Choristoneura rosaceana*) are involved (Table 2).

### 3.5. The RNA-Sequencing Expression of RALF-Like Genes in Strawberry and Apple

The downloaded RNA-seq data were used to analyze the expression profiles of *RALF-*like genes and the basic information of these datas listed in Appendix A. In wild strawberry, *Fv-RALF1*, *Fv-RALF2*, *Fv-RALF3*, *Fv-RALF5*, and *Fv-RALF12* were more highly expressed than other genes (Appendix A). *Fv-RALF1* shows a lower expression level in the ghost development stages and wall 4, wall 5 stages, which means it may affect the development of ghost and wall. *Fv-RALF2* and *Fv-RALF3* all have different expression profiles in the cortex compared with other tissues, which implies the potential functions on cortex development. *Fv-RALF5* and *Fv-RALF12* both have lower expression in embryo than other tissues and may influence the development of the embryo. *Fv-RALF4*, *Fv-RALF6*, *Fv-RALF7*, *Fv-RALF8*, *Fv-RALF9*, *Fv-RALF10*, and *Fv-RALF11* all have higher expression levels in specific tissues (Appendix A). *Fv-RALF4*, *Fv-RALF9*, and *Fv-RALF10* are more highly expressed pollen-specific genes. These genes may play an important role in the development of pollen. The receptacle fruit at green stage (15 days post-anthesis) and white (turning) stage (22 days post-anthesis) with achenes removed were used to analyze the *RALF*-like gene expression profile (Appendix A). As depicted, *Fv-RALF1*, *Fv-RALF5*, *Fv-RALF12*, and *Fv-RALF13* all show higher expression levels in some ways. To retrieve the expression profile of *RALF*-like genes in achene and receptacle fruit, the RNA-seq data of one red-fruited and two natural white-fruited strawberry varieties in two tissues and three ripening stages were used (Appendix A). *Fv-RALF1*, *Fv-RALF2*, *Fv-RALF5*, *Fv-RALF10*, and *Fv-RALF12* were more highly expressed in some cultivars and tissues, and showed variable expression profiles, which means they may have an effect on the development of achene and receptacle fruit. The powdery mildew-infected leaf sample from Hawaii was used to analyze the expression level under fungal stress (Appendix A). *Fv-RALF12* always showed a high expression levels, though this does not exclude it from having an essential function in resistance from infection by powdery mildew. However, *Fv-RALF1* is more highly expressed one day after infection, which may represent a response to infection.

In cultivated strawberry, the RNA-seq data of the development achene and receptacle fruit were also analyzed (Appendix A). *Fa-RALF1*, *Fa-RALF18*, and *Fa-RALF39* are more highly expressed at the white receptacle stage, whereas, *Fa-RALF14*, *Fa-RALF24*, and *Fa-RALF33* exhibit a higher level of expression in leaf than other tissues, and *Fa-RALF29* and *Fa-RALF35* may be highly expressed root-specific genes. In diverse development stages of cultivated strawberry (Appendix A), we reported that *Fa-RALF1*, *Fa-RALF10*, and *Fa-RALF33* were all more highly expressed at the turning stage compared to other stages. *Fa-RALF6*, *Fa-RALF17*, *Fa-RALF26*, and *Fa-RALF32* were more highly expressed at the green fruit stage.

According to the RNA-seq data of apple, we reported that Md-RALF7 and Md-RALF8 may have an effect on the development of flower (Appendix A). *Md-RALF3*, *Md-RALF4*, *Md-RALF13*, and *Md-RALF16* are more highly expressed in root tip (Appendix A). *Md-RALF3*, *Md-RALF4*, and *Md-RALF13* were more highly expressed than other isotypes in during infection by *Alternaria alternata* (Appendix A). *Md-RALF3* responds to infection after 12 h. *Md-RALF4* responds to the infection after 18 h. The expression level of *Md-RALF13* remained high until 72 h, in which it began to decline after.

### 3.6. RT-qPCR Analysis of Nineteen RALF-Like Genes in Cultivated Strawberry Related to Hormones

Nineteen RALF-like genes of cultivated strawberry under different hormone stresses were analyzed by RT-qPCR (Figure 6). Herein, we reported that *Fa-RALF1* (GA), *Fa-RALF2* (ABA), *Fa-RALF5* (NAA), *Fa-RALF8* (NAA), *Fa-RALF10* (SA), *Fa-RALF12* (GA), *Fa-RALF13* (ABA), *Fa-RALF14* (ABA and SA), *Fa-RALF18* (MeJA), *Fa-RALF31* (ABA and SA), *Fa-RALF32* (ABA), *Fa-RALF33* (MeJA), *Fa-RALF35* (MeJA), and *Fa-RALF39* (ABA) have an obvious response to one or two hormones, whereas *Fa-RALF19*, *Fa-RALF24*, *Fa-RALF25*, *Fa-RALF29*, and *Fa-RALF40* evidently respond to all the stresses, which implies a role for RALF genes in hormone response. Here, fourteen genes (*Fa-RALF1*, *Fa-RALF2*, *Fa-RALF5*, *Fa-RALF10*, *Fa-RALF12*, *Fa-RALF13*, *Fa-RALF14*, *Fa-RALF19*, *Fa-RALF24*, *Fa-RALF29*, *Fa-RALF32*, *Fa-RALF35*, *Fa-RALF39*, and *Fa-RALF40*) are mature *RALF* genes and seven genes (*Fa-RALF5*, *Fa-RALF10*, *Fa-RALF14*, *Fa-RALF19*, *Fa-RALF24*, *Fa-RALF29*, and *Fa-RALF35*) have coding capacity.

## 4. Discussion

### 4.1. Expression Profile of RALF-Like Genes in Seven Rosaceae Species

As reported, RALF genes are not only involved in several biotic and abiotic stresses but also influence the growth and development of plants. Here, we found that the expression profiles are mostly similar with the researchers reported, such as the drought response [13], root growth [64], hormone response [18], fungi response [18,19,20], cell wall development [15,17], and the flower development (especially the development of pollen tube) [14,16]. In the present study, we combined the results of RNA-seq, EST data, cis-regulatory elements analysis, and RT-qPCR to identify several genes which have identical expression profiles in two or more analyses, which may give some instructions of the research of RALF-like genes among Rosaceae species.

*Fv-RALF10* exhibits drought responses according to the analysis of EST data and cis-regulatory elements. RNA-seq consistent with cis-regulatory element analysis involved in *Fv-RALF1*, *Fv-RALF2*, *Fv-RALF12*, *Fv-RALF15* (seed, embyro, and endosperm), which have no research reported, may be the special findings in our study among Rosaceae species. The RALF genes most reported were the role they play in root growth and pollen tube development [14,16]. Here, the comparison between RNA-seq and EST data reveal that *Md-RALF3* may have influence on the growth of young root, and *Md-RALF7* and *Md-RALF8* may affect the development of flowers. As reported that MeJA may influence the expression of RALF genes [18]; here, we tried more hormones, which may influence the tissue development and stresses response, the results from RT-qPCR and cis-regulatory elements showed that *Fa-RALF1*, *Fa-RALF2*, *Fa-RALF12*, *Fa-RALF13*, *Fa-RALF14*, *Fa-RALF18*, *Fa-RALF19*, *Fa-RALF24*, *Fa-RALF25*, *Fa-RALF29*, *Fa-RALF31*, *Fa-RALF33*, and *Fa-RALF40* have identical expression profiles, in other words, they may have influence on the growth, development and stresses response of Rosaceae species.

In strawberry, the *FaRALF-33-like* gene, which is homologous to *FvRALF-33-like* (GenBank accession No.: XM_011460413.1) and *AtRALF33* (named *At-RALF1* here), responded to infection by three different fungi [20]. Local BLASTP [46] was used to search the similar genes among our 163 sequences, using FvRALF-33-like and AtRALF33 as the query sequences. We revealed that Fa-RALF1 (GenBank accession No.: XP_011458715) and Fv-RALF5 are the best-match sequences. As shown (Figure 7), except for Fa-RALF1, other sequences all possess the typical features and the integral YISY, YYNC, RCR motifs.

Fv-RALF5 belongs to clade II, and among the clade, there are seventeen other mature RALF genes (Pc-RAL2, Md-RALF4, Pc-RALF6, Md-RALF11, Pa-RALF10, Pm-RALF5, Pp-RALF5, Fa-RALF35, Fa-RALF29, Fa-RALF5, Pp-RALF6, Pm-RALF7, Pa-RALF1, Pc-RALF7, Md-RALF7, Md-RALF12, and Fa-RALF10) from Rosaceae species, and no the RCR motif was found in Pc-RALF6. Therefore, the 16 *RALF-like* genes, except for *Pc-RALF6*, may have similar functions as *Fv-RALF5*, *At-RALF1*, and *FaRALF-33-like* genes. As depicted in the phylogenetic tree, we reported that mature RALF genes in clade II, except Pc-RALF6, At-RALF4, and At-RALF1, were similarly divided within the same clade, which is consistent with previous reports. Stegmann et al. [65] already investigated that *RALF33* (AT4G15800, named *At-RALF1* here), *RALF23* (AT3G16570, named *At-RALF4* here), and *RALF34* (AT5G67070, named *At-RALF14* here) negatively regulate immunity, whereas *RALF32* (AT4G14010, named *At-RALF5* here) did not. Fv-RALF5 clustered with Fa-RALF10, which implies that they have a similar function. They may also respond to the fungal infection. Furthermore, according to the RNA-seq results and cis-regulatory element analysis, we reveal that they may have influence on the development of receptacle and be involved in some hormone stresses (ABA, MeJA, etc.) and low temperature stress. Compared with the expression profiles of *Arabidopsis* in Campbell and Turner [4], we found that the clade IV (clade I-A and clade III-A concluded here) are not consistent with our research mostly. The expression profiles are involved in hormone, fruit development, flower development, leaf growth, salt stress, powdery mildew response, and achene development.

Ten homologous genes pairs were selected from the phylogenetic trees of mature peptides (Md-RALF4 and Pc-RALF2; Pp-RALF5, Pm-RALF5, and Pa-RALF10; Fa-RALF5, Fa-RALF29, and Fa-RALF35; Pm-RALF7 and Pp-RALF6; Fa-RALF2, Fv-RALF2, and Fa-RALF40; Md-RALF14 and Md-RALF5; Pp-RALF12 and Pa-RALF3; Pm-RALF6, Pp-RALF8, and Pa-RALF2; Pc-RALF9 and Md-RALF15; Fa-RALF19, Fv-RALF1, Fa-RALF33, Fa-RALF24, and Fa-RALF14), and except for Fv-RALF5 and Fa-RALF10, which were analyzed above, we examined whether these gene pairs had similar expression profiles using RNA-seq, EST, and RT-qPCR combined with cis-regulatory element analysis. The integrative analysis of RT-qPCR and cis-regulatory element implied that the *Fa-RALF5*, *Fa-RALF29*, and *Fa-RALF35* may respond to auxin stress, *Fa-RALF2*, *Fv-RALF2*, and *Fa-RALF40* to ABA, and *Fa-RALF19*, *Fv-RALF1*, *Fa-RALF33*, *Fa-RALF24*, and *Fa-RALF14* to auxin and ABA. Based on the RNA-seq data, *Fa-RALF2*, *Fv-RALF2*, and *Fa-RALF40* may affect the development of cortex, achene, and receptacle fruit; *Fa-RALF*19, *Fv-RALF1*, *Fa-RALF33*, *Fa-RALF24*, and *Fa-RALF14* were mainly expressed in leaf and may also be expressed in achene, receptacle fruit, ghost, wall and may all be involved in response to fungal infection. As for the EST data, we reported that *Pp-RALF5*, *Pm-RALF5*, and *Pa-RALF10* are expressed in leaf; *Fa-RALF5*, *Fa-RALF29*, and *Fa-RALF35* in red fruit; *Pm-RALF7* and *Pp-RALF6* in mesocarp; *Pp-RALF12* and *Pa-RALF3* in fruit; *Pm-RALF6*, *Pp-RALF8*, and *Pa-RALF2* in mesocarp; *Md-RALF14* and *Md-RALF5* in fruit and flower; and *Pc-RALF9* and *Md-RALF15* in fruit, flower, young shoots, and leaf. Notably, *Md-RALF14* and *Md-RALF5* expression is also altered upon fungal infection.

The genes mentioned above are essential to the growth, development, and stress response. In particular, the mature RALF genes were already found to exhibit obvious expression differences. In our research, there are 24 mature RALF genes with coding capacity (*Fv-RALF5*, *Fv-RALF1*, *Fa-RALF19*, *Fa-RALF33*, *Fa-RALF14*, *Fa-RALF24*, *Md-RALF2*, *Md-RALF4*, *Md-RALF15*, *Md-RALF11*, *Md-RALF12*, *Fa-RALF35*, *Pa-RALF1*, *Pa-RALF10*, *Pc-RALF1*, *Pc-RALF2*, *Pc-RALF6*, *Pc-RALF9*, *Pp-RALF5*, *Pp-RALF6*, *Pm-RALF7*, *Pp-RALF8*, *Pm-RALF6*, and *Pm-RALF5*). As mentioned above, these 24 coding mature RALF genes are widely expressed, and 18 genes (*Fv-RALF1*, *Fa-RALF14*, *Fa-RALF19*, *Fa-RALF24*, *Fa-RALF33*, *Fa-RALF35*, *Md-RALF2*, *Md-RALF11*, *Md-RALF12*, *Md-RALF15*, *Pa-RALF10*, *Pc-RALF9*, *Pm-RALF5*, *Pm-RALF6*, *Pm-RALF7*, *Pp-RALF5*, *Pp-RALF6*, and *Pp-RALF8*) may be involved in the development of tissues and organs, which include 12 tissues or organs. In particular, seven genes have a tissue- or organ-specific expression profile. For example, *Pa-RALF10*, *Pp-RALF5*, and *Pm-RALF5* may be leaf-specific genes, whereas *Pm-RALF6*, *Pm-RALF7*, *Pp-RALF6*, and *Pp-RALF8* showed mesocarp-specific expression. Meanwhile, five genes (*Fa-RALF35*, *Fa-RALF19*, *Fa-RALF33*, *Fa-RALF14*, and *Fa-RALF24*) may be involved in hormone response. Notably, there are three genes (*Fv-RALF5*, *Fa-RALF33*, and *Md-RALF14*) related to fungal infection.

The GO terms give a clear function cluster of RALF-like genes, which involved in the pollen tube development, root development, negative regulation of growth, cell wall growth, hormone activaty, brassinosteroid-mediated signaling pathway, and so on. According to the classification, we found they are very consistent with previous researches about the functions of RALF genes [1,14,15,16,17,18,64]. Fungal infection response and pollen tube development are important for Rosaceae species, and the RALF-like gene expression in seed and fruit among Rosaceae species specially.

### 4.2. Classification, Duplication Events and Evolution of RALF-Like Proteins in Seven Rosaceae Species

In several researches, the classification and evolution of RALF-like proteins were analyzed. Cao et al. [9] characterized RALF proteins in *Arabidopsis*, rice, poplar, and maize, which were divided into ten groups, and elucidated that tandem duplication played a dominant role in the expansion of RALF gene family and RALF-like proteins mainly through purifying selection, which is consistent with the conclusions of our research. However, we found that the dominant duplication event was WGD or segmental, not tandem. Sharma et al. [11] divided RALF-like proteins of *Arabidopsis*, rice, maize, and soybean into seven groups, and the RALF-like genes in 51 plant species also diverged into four major clades [4]. In our work, four clades were also divided among RALF-like genes between *Arabidopsis* and seven Rosaceae species.

The number of RALF-like genes was not consistent with the previous. Cao et al. [9] identified 33 RALF-like genes in *Arabidopsis*; Sharma et al. [11] identified 39; Campbell and Turner [4] identified 37; and, in our research, we retrieved 39 from analysis. Due to the reference genomes we selected not being identical with Campbell and Turner [4], we identified 13, and not 9, RALF-like genes in *Fragaria vesca*, 20, and not 33, in *Malus* × *domestica*, and 14, and not 13, in *Prunus persica*. Meanwhile, *Fragaria* × *ananassa* and not *Arabidopsis* had the most RALF-like genes (41). The classification in our study among RALF-like genes in *Arabidopsis* largely confirmed previous reports.

The alignment of RALF-like genes was implemented using Jalview Version 2 (muscle with defaults) [43], and the residue conserved within the mature peptide region of the four major clades was demonstrated using WebLogo 3 [66]. As depicted in Appendix A compared with the analysis of Campbell and Turner [4], clade I–C, clade I–G, clade II, and clade IV were similar to clade I and II (A), clade I–D, clade I–E, clade I–F to clade III, clade I–A to clade IV, clade III–B to clade II (B), and clade III–A to clade IV, whereas clade I-B did not share any similarity to other clades and has none of the features of RALF genes. Campbell and Turner [4] reported that the genes in clade IV were not mature RALF genes. They also reported the RALF-like genes in clade I–A and clade III–A to be RNA-like genes, which is consistent with our research.

## 5. Conclusions

This study provides extensive information on the RALF-like genes of seven Rosaceae species, and involved, including their identification, an analysis of their coding capacity, evolutionary relationships, duplication events, motifs, domains, GO annotation, cis-regulatory elements, and EST data, as well as use of RNA-seq and RT-qPCR to characterize expression profiles. These RALF-like genes are divided into four clades, and purifying selection plays a main role in the evolutionary process. In particular, the 24 RALF genes coding for mature proteins among the Rosaceae species deserve to be further researched in regard to hormone response, fungal infection, growth, and development.

## Figures and Tables

**Figure 1 genes-11-00174-f001:**
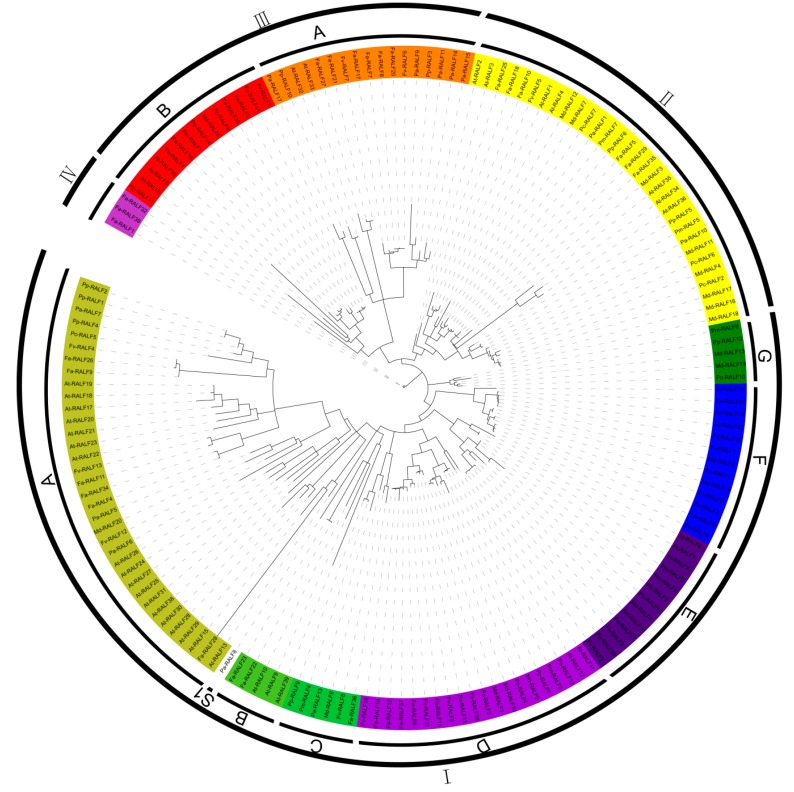
The maximum-likelihood phylogenetic tree of RALF-like proteins among *Arabidopsis* and seven Rosaceae species.

**Figure 2 genes-11-00174-f002:**
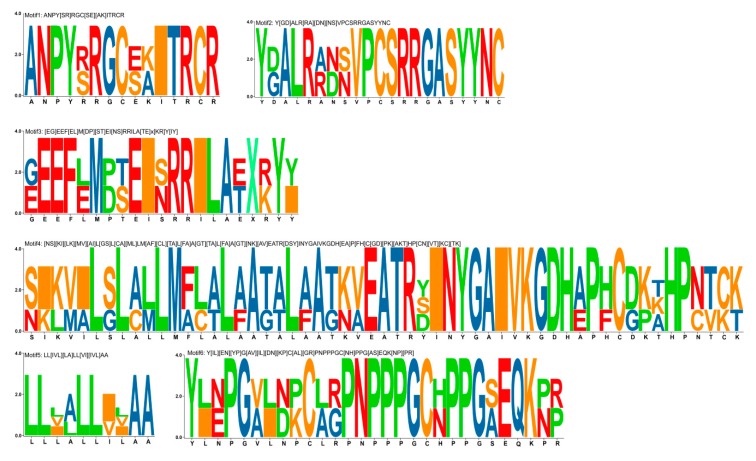
The six motifs in the research.

**Figure 3 genes-11-00174-f003:**
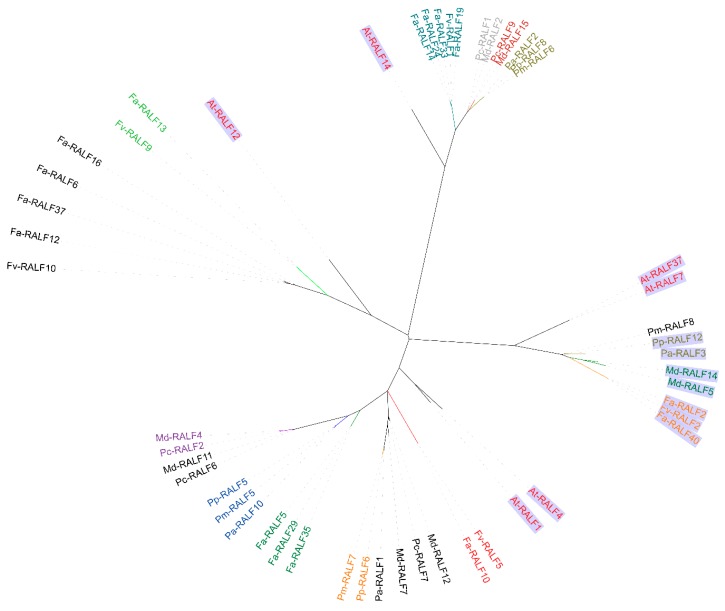
The maximum-likelihood phylogenetic tree of 51 mature RALF genes examined in the present study. RALF genes from *Arabidopsis* marked with the same color and background, and the same gene pair with the same mark.

**Figure 4 genes-11-00174-f004:**
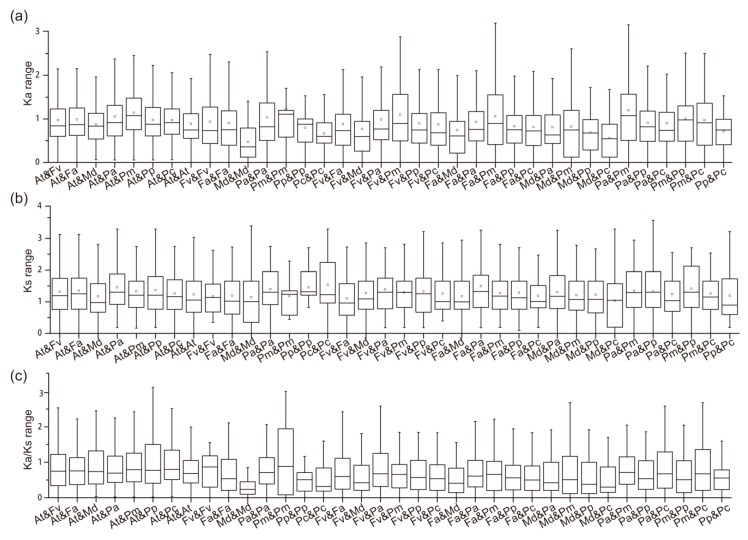
The ka, ks, and ka/ks value distributions. (**a**) ka values; (**b**) ks values; (**c**) ka/ks values.

**Figure 5 genes-11-00174-f005:**
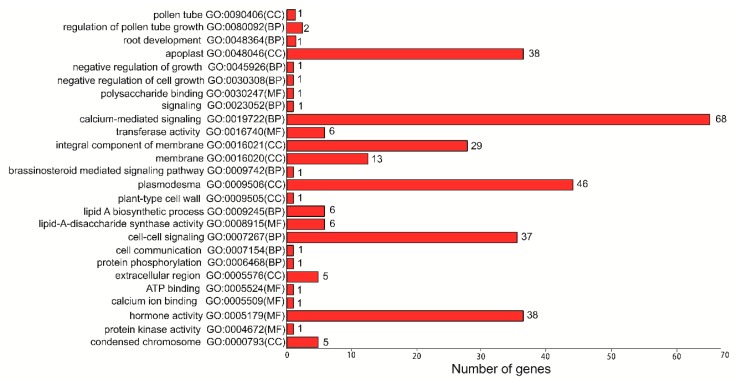
The gene ontology (GO) term distribution and functional clustering among seven Rosaceae species.

**Figure 6 genes-11-00174-f006:**
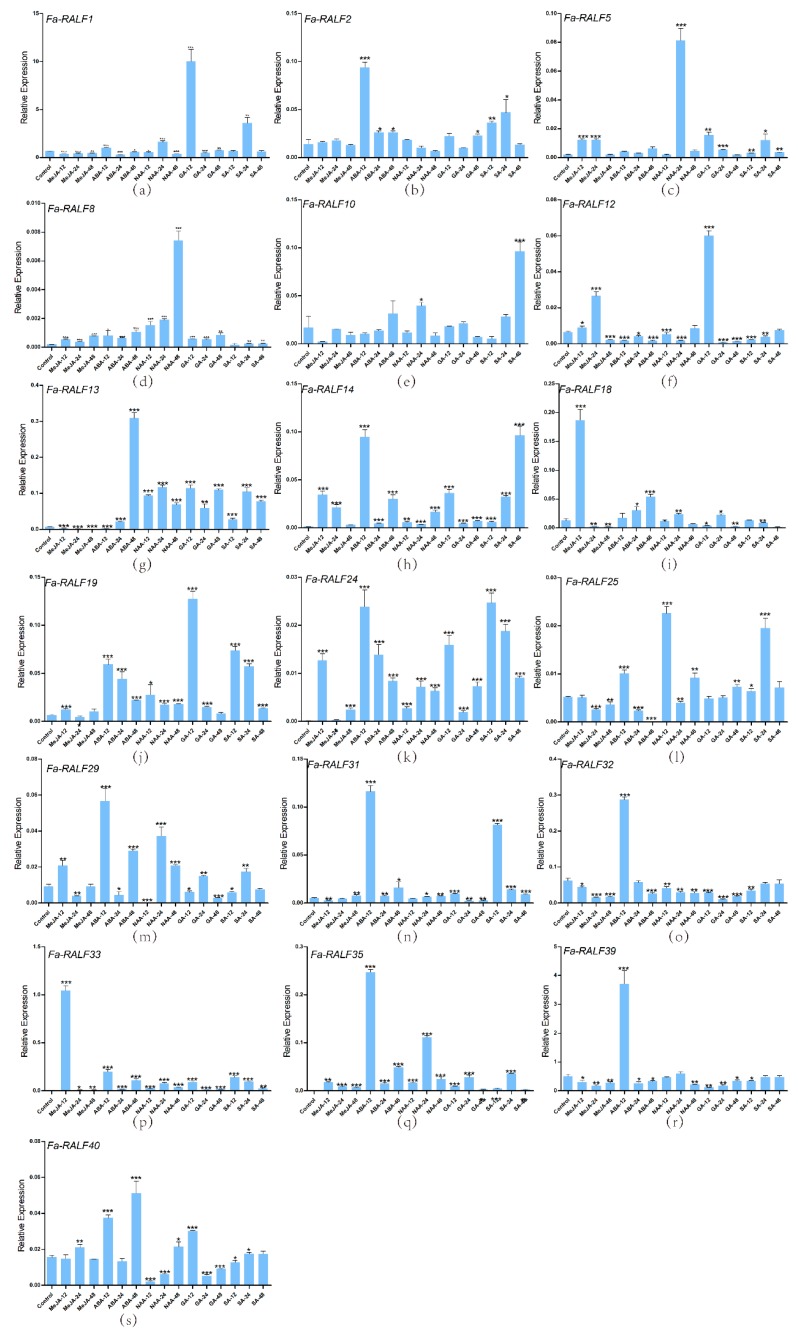
Relative expression profiles of 19 RALF-like genes from *Fragaria* × *ananassa* under different hormone treatments (**a**–**s**). Error bars represent standard deviation (SD). Columns with stars marked the significantly different. The numbers of stars stand for the difference extent. The more stars, the higher the significant difference; three stars represents an extreme difference. Here, all the treatments are compared with “control”, and * represents *p* ≤ 0.05, ** represents *p* ≤ 0.01, *** represents *p* ≤ 0.001. MeJA-12,-24,-48: the leaves treated by MeJA after 12, 24, and 48 h; ABA-12,-24,-48: the leaves treated by ABA after 12, 24, and 48 h; NAA-12,-24,-48: the leaves treated by NAA after 12, 24, and 48 h; GA-12,-24,-48: the leaves treated by GA after 12, 24, and 48 h; SA-12,-24,-48: the leaves treated by SA after 12, 24, and 48 h.

**Figure 7 genes-11-00174-f007:**
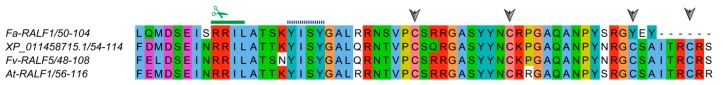
The alignment of three RALF genes similar to FaRALF-33-like.

**Table 1 genes-11-00174-t001:** Primers used to perform quantitative real-time PCR (RT-qPCR) for *RALF-*like genes.

Symbol	Gene ID	Forward Sequence (5′ to 3′)	Reverse Sequence (5′ to 3′)
Fa-RALF1	snap_masked-Fvb4-2-processed-gene-95.31-mRNA-1	CAACCTCCTCATTCTCCTCAC	TGCTGGTGGCTAAGATGC
Fa-RALF2	augustus_masked-Fvb4-2-processed-gene-134.10-mRNA-1	TCCTCACCCATTTCTCAATC	CCTTCTTCAGTGTCTCGTAGC
Fa-RALF5	snap_masked-Fvb2-1-processed-gene-164.41-mRNA-1	ACGGGTTGAGCTTTGTTCCT	GGCCAAGATACGCCTGTTGA
Fa-RALF8	snap_masked-Fvb6-1-processed-gene-108.14-mRNA-1	CATGGAGGCGCAATGTCAAG	AATAATTCTGGCAGTCACCGGA
Fa-RALF10	augustus_masked-Fvb6-1-processed-gene-322.10-mRNA-1	AGCTGGTGGAGACCTCTCAT	CGGTTATAGGGGTTGGCCTG
Fa-RALF12	augustus_masked-Fvb3-3-processed-gene-31.4-mRNA-1	ACTTCTTCTGTTTGTCCACCC	GACGACTCTACTGCAGCCAC
Fa-RALF13	augustus_masked-Fvb3-3-processed-gene-31.5-mRNA-1	CATCAGCTACGGTGCCCTAA	CTGCAACCTCGACGATAGGT
Fa-RALF14	augustus_masked-Fvb1-1-processed-gene-196.6-mRNA-1	TTCTTACGGTGCGCTCTCAG	GCAGCCTCTGCTGTAAGGAT
Fa-RALF18	augustus_masked-Fvb6-3-processed-gene-378.1-mRNA-1	TGGATACCCACCACCGTTTC	GCTATAGGGGTTGGCCTGAG
Fa-RALF19	augustus_masked-Fvb1-4-processed-gene-81.5-mRNA-1	TGACTTCTTCAACTCTACTCCACTA	GCCCTGGCGGAAGGGCAGTCAGGGC
Fa-RALF24	augustus_masked-Fvb1-3-processed-gene-90.5-mRNA-1	GCTTCTCCAACTCTACTCCACTATC	CGACTCGTCGAGATGGGCCCTGGCG
Fa-RALF25	snap_masked-Fvb6-4-processed-gene-50.20-mRNA-1	TCTCCGGCTTAGTCTCCGAC	GGCTATAGGGGTTGGCCTGA
Fa-RALF29	augustus_masked-Fvb2-3-processed-gene-83.11-mRNA-1	CATTGCAGAGTGCATGGCTG	GTAGGGATTAGCCTGTGCCC
Fa-RALF31	snap_masked-Fvb4-3-processed-gene-153.15-mRNA-1	TAATCTTCTACCTGGGTCTCCTCTT	TTTGAGTGCATTGAGGTCCAGAACT
FXARAL32	maker-Fvb4-3-snap-gene-192.46-mRNA-1	GCAATGGGTCTTCAGTTCTG	GCAATGGGTCTTCAGTTCTG
Fa-RALF33	augustus_masked-Fvb1-2-processed-gene-103.4-mRNA-1	ATGACTTCTTCAACTCTACTCCACT	AACTTCACGCTCGCCTCGTCGAGCT
Fa-RALF35	snap_masked-Fvb2-2-processed-gene-47.50-mRNA-1	GGCAAAGTCCTCTTCCATTATTCTC	CGGGACTTGGCCGGAACAAAGCTCA
Fa-RALF39	snap_masked-Fvb4-4-processed-gene-106.36-mRNA-1	TCCTCATTCTCCTCACTACTACTCT	GGAAGCCGTGGAATGGGCGGCCAGG
Fa-RALF40	augustus_masked-Fvb4-4-processed-gene-145.10-mRNA-1	AAGGTCGGGGACTGCATAAC	TCGTGTAATGACCTCACAGCC
EF-1α	XM_004307362	CATGCGCCAGACTGTTGCTGT	GACCGACTCAGAATACTAGTAGC

**Table 2 genes-11-00174-t002:** The expression profile of RALF-like genes from seven Rosaceae species from EST data.

Plant Tissues and Stresses	Genes Retrieved from Expression Sequence Tag (EST) Data
Sault	*Fv-RALF10, Fv-RALF11, Fv-RALF12*
Drought	*Fv-RALF10*
Heat	*Fv-RALF11, Fv-RALF12*
Red fruit/Fruit	*Fa-RALF5, Fa-RALF29, Fa-RALF35, Md-RALF2, Md-RALF3, Md-RALF 5, Md-RALF6, Md-RALF7, Md-RALF11, Md-RALF12, Md-RALF13, Md-RALF14, Md-RALF15, Md-RALF19, Md-RALF20, Pp-RALF12, Pp-RALF13*
Flower	*Md-RALF1, Md-RALF9, Md-RALF10, Md-RALF2, Md-RALF5, Md-RALF7, Md-RALF11, Md-RALF12, Md-RALF13, Md-RALF14, Md-RALF 15, Md-RALF 19, Md-RALF 20*
Bud	*Md-RALF2, Md-RALF3, Md-RALF7, Md-RALF11, Md-RALF13, Md-RALF 15, Md-RALF 19, Md-RALF 20*
Young root	*Md-RALF3, Md-RALF7, Md-RALF11*
Xylem	*Md-RALF 6, Md-RALF 7, Md-RALF13, Md-RALF19, Md-RALF20*
Young shoot	*Md-RALF2, Md-RALF7, Md-RALF13, Md-RALF15, Md-RALF19, Md-RALF 20*
Leaf	*Md-RALF2, Md-RALF13, Md-RALF15, Md-RALF19, Md-RALF20, Md-RALF, Pp-RALF5*
Shoot internodes	*Md-RALF 13, Md-RALF 19*
Fruit cortex	*Md-RALF 13, Md-RALF 19*
Fruit core	*Md-RALF 13, Md-RALF 19*
Phloem	*Md-RALF 20*
Mesocarp	*Pp-RALF6, Pp-RALF8, Pp-RALF13*
*Venturia inaequalis*	*Md-RALF5, Md-RALF14*
*Choristoneura rosaceana*	*Md-RALF20*

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
