# Peer review of "The Genome-Wide Analysis of RALF-Like Genes in Strawberry (Wild and Cultivated) and Five Other Plant Species (Rosaceae)"

_genes, 2020, doi:10.3390/genes11020174_

Round 1
Reviewer 1 Report
The authors identified 124 RALF-like genes across seven Rosaceae species and conducted subsequent functional and structural analysis on these genes. I have several suggestions to improve the manuscript:
1. In phylogenetic analysis (Figures 1), the authors only included Arabidopsis RALF genes and concluded most RALF-proteins have conserved functional structures. I would like to see if the conclusion still holds by adding more RALF sequences from Campbell and Turner 2017, who reported 765 RALF genes across 51 species. The same set of outgroup RALF genes should be used in Figure 5, to test if homologous gene pairs still hold.
2. In RNA-seq analysis, what is the rationale to choose the three species presented in the paper? Because these are the only ones with RNA-seq data in different developmental stages, tissue types, etc? If so, the author should make such statement in the paper. In the meantime, a supplementary table should be given to present such information as to which species has what kind of dataset (stage, tissue, stress). Moreover, I found some datasets are based on paired-end sequencing (e.g., ERP004230, ERP013896, SRP111905, SRP139480), how the authors could use RPKM to represent the expression level for paired-end data?
3. The GO analysis result part is confusing. There are 124 RALF-like genes identified in the paper, but the authors claimed that "51% (138/312) of RALF-like genes were enriched...", how is this possible? Did you mean GO terms? Do we have similar GO terms in our study comparing to others? What is the difference? I didn't see much discussion. The same applied to RNA-seq results. For examples, are the same genes possibly involved in hormone response also found in other non-Rosaceae species? Are there any unique expression patterns found in Rosaceae species? Also, do the genes in four clades found in the study have similar/unique functions as revealed in Campbell and Turner 2017? I feel Discussion is weak and not deep enough.
4. Minor points: line 19, change "evolution" to "evolutionary"; line 47, remove "Several"; line 54, specify "developmental stages" rather than just "stages"; delete lines 91-93; lines 103-105, I feel the RNA-seq datasets were downloaded from NCBI SRA, the authors need to double check this statement; Too many figures in the main text, I suggest put Figures 2, 3, 6, 8-9, 11-13 into supplementary figures.
Author Response
Dear reviewer,
I have already adjusted the content in my manuscript, and i will submit the response as a text, if there are some questions, please give your suggestions and introductions, thank you so much!
Sincerely,
Hong Zhang

Reviewer 2 Report
Please see the attached file with my detailed feedback.

Author Response

(The authors gave the same response as above.)

Round 2
Reviewer 1 Report
In the response letter, the authors stated:
"the RNA-seq analysis process I used like this below:
Single-end: Bowtie2--Samtools--Stringtie
Paired-end: Hisat2--Samtools--Stringtie"
Therefore, the authors must have used both RPKM and FPKM. Such information was not presented. Please add as the current writing is misleading.
Also, the authors should check the spellings throughout the entire manuscript to make sure words are spelling correctly. For example, "homone response" (line 388) should be "hormone response".
Author Response
Dear reviewer,
Thank you so much for the suggestion about RPKM and FPKM, and I could explain that the values of RPKM and FPKM are the same in single-end RNA-seq datas. And the results of stringtie here are all FPKM.
Meanwhile, I have already checked the spellings throughout the entire manuscript and corrected them.
Kind regards,
Hong Zhang